# Salicylate, diflunisal and their metabolites inhibit CBP/p300 and exhibit anticancer activity

Kotaro Shirakawa[1,2,3,4], Lan Wang[5,6], Na Man[5,6], Jasna Maksimoska[7,8], Alexander W Sorum[9], Hyung W Lim[1,2], Intelly S Lee[1,2], Tadahiro Shimazu[1,2], John C Newman[1,2], Sebastian Schröder[1,2], Melanie Ott[1,2], Ronen Marmorstein[7,8], Jordan Meier[9], Stephen Nimer[5,6], Eric Verdin[1,2]*

[1]Gladstone Institutes, University of California, San Francisco, United States; [2]Department of Medicine, University of California, San Francisco, United States; [3]Department of Hematology and Oncology, Kyoto University, Kyoto, Japan; [4]Graduate School of Medicine, Kyoto University, Kyoto, Japan; [5]University of Miami, Gables, United States; [6]Sylvester Comprehensive Cancer Center, Miami, United States; [7]Perelman School of Medicine, University of Pennsylvania, Philadelphia, United States; [8]Department of Biochemistry and Biophysics, Abramson Family Cancer Research Institute, Philadelphia, United States; [9]Chemical Biology Laboratory, National Cancer Institute, Frederick, United States

**Abstract** Salicylate and acetylsalicylic acid are potent and widely used anti-inflammatory drugs. They are thought to exert their therapeutic effects through multiple mechanisms, including the inhibition of cyclo-oxygenases, modulation of NF-κB activity, and direct activation of AMPK. However, the full spectrum of their activities is incompletely understood. Here we show that salicylate specifically inhibits CBP and p300 lysine acetyltransferase activity *in vitro* by direct competition with acetyl-Coenzyme A at the catalytic site. We used a chemical structure-similarity search to identify another anti-inflammatory drug, diflunisal, that inhibits p300 more potently than salicylate. At concentrations attainable in human plasma after oral administration, both salicylate and diflunisal blocked the acetylation of lysine residues on histone and non-histone proteins in cells. Finally, we found that diflunisal suppressed the growth of p300-dependent leukemia cell lines expressing AML1-ETO fusion protein *in vitro* and *in vivo*. These results highlight a novel epigenetic regulatory mechanism of action for salicylate and derivative drugs.

*For correspondence: everdin@ gladstone.ucsf.edu

**Competing interests:** The authors declare that no competing interests exist.

## Introduction

The anti-inflammatory activity of salicylate was first described by the Greek physician Hippocrates. One of its widely used derivatives, acetylsalicylic acid (Aspirin), inhibits prostaglandin biosynthesis by irreversibly inactivating cyclooxygenases via non-enzymatic acetylation of a single serine residue (*Warner et al., 1999*). Interestingly, salicylic acid does not possess this acetylating activity (since it is lacking the acetyl group) and does not inhibit cyclooxygenase *in vitro*.

However, salicylic acid blocks cyclooxygenase expression at the transcriptional level thereby explaining its anti-inflammatory properties (*Xu et al., 1999*). In addition, both salicylic acid and aspirin inhibit nuclear factor kappa B (NF-κB) activity (*Kopp and Ghosh, 1994*) by inhibiting IκB kinase β (IKKβ) (*Yin et al., 1998*). Other possible mechanisms of action have been proposed that include JNK pathway inhibition (*Schwenger et al., 1997*) and direct allosteric activation of AMP kinase (AMPK)

**eLife digest** People have been using a chemical called salicylate, which was once extracted from willow tree bark, as medicine for pain, fever and inflammation since ancient Greece. Aspirin is derived from salicylate but is a more potent drug. Aspirin exerts its anti-inflammatory effect by shutting down the activity of proteins that would otherwise boost inflammation. Aspirin achieves this by releasing a chemical marker, called an acetyl group, to be added to these proteins via a process known as protein acetylation. However, salicylate cannot trigger protein acetylation and so it was not clear how it reduces inflammation.

An anti-diabetes drug that is converted into salicylate in the body reduces inflammation by inhibiting a protein called NF-κB. In 2001, a group of researchers reported that NF-κB becomes active when an enzyme called p300 adds an acetyl group to it. This raised the question: does salicylate reduce inflammation by blocking, instead of triggering, protein acetylation.

Now, Shirakawa et al. – who include a researcher involved in the 2001 study – show that salicylate does indeed block the activity of the p300 enzyme. Shirakawa et al. then searched a database looking for drugs that have salicylate as part of their molecular structure. The search led to a drug called diflunisal, which was even more effective at blocking p300 in laboratory tests.

Some cancers, including a blood cancer, rely on p300 to grow; diflunisal was shown to stop this kind of cancer cell from growing, both in the laboratory and in mice. Together, the experiments suggest that salicylate and drugs that share some of its structure might represent useful treatments for certain cancers, as well as other diseases that involve the p300 enzyme.

(*Hawley et al., 2012*). However, the pleiotropic effects of salicylate treatment on different cell types remain incompletely understood.

Salsalate, a salicylate precursor, is an effective therapy for type 2 diabetes (*Goldfine et al., 2010*), a metabolic disorder associated with insulin resistance and a strong pro-inflammatory component dependent on NF-κB (*Donath and Shoelson, 2011*). The efficacy of salicylates on insulin resistance is thought to reflect its anti-inflammatory activity and to be mediated by IKKβ inhibition (*Yin et al., 1998*).

Interestingly, examination of the chemical structure of anacardic acid, a previously reported p300 inhibitor, revealed that it contained a salicylic acid moiety linked to a long alkyl chain (*Balasubramanyam et al., 2003*; *Sung et al., 2008*). Previously, we showed that full activation of NF-κB activity requires the reversible acetylation of NF-κB by CBP/p300 histone acetyltransferases (HATs) (*Chen et al., 2001*). Here, we have tested the possibility that salicylic acid might exert its transcriptional inhibitory activity by directly affecting NF-κB acetylation via inhibition of CBP/p300 acetyltransferase activity.

## Results

### Salicylate inhibits CBP/p300 acetyltransferase activity by directly competing with acetyl-CoA *in vitro*

To determine whether salicylate inhibits p300 and other acetyltransferases, we used *in vitro* acetylation assays with purified histones and a recombinant p300 catalytic domain. Salicylate effectively inhibited p300 dependent acetyltransferase activity ($IC_{50}$ = 10.2 mM) and CBP-mediated acetyltransferase activity ($IC_{50}$ = 5.7 mM) *in vitro*, but did not detectably inhibit PCAF or GCN5 acetyltransferases (*Figure 1A*) *in vitro*.

To confirm that salicylate binds to p300, we used thermal stability assays. A p300 HAT domain construct (residues 1279–1666) bearing an inactivating Tyr1467Phe mutation to facilitate purification of homogeneously hypoacetylated p300 was expressed and purified with an N-terminal 6-His tag from *E. coli* cells. The protein was further purified by chromatography and incubated with increasing concentrations of sodium salicylate for 30 min and with SYPRO orange dye (Invitrogen). Thermal melt curves were obtained by heating the protein from 20–95°C and monitoring fluorescence at 590 nm. This experiment revealed that the thermal unfolding temperature of p300/acetyl-CoA was

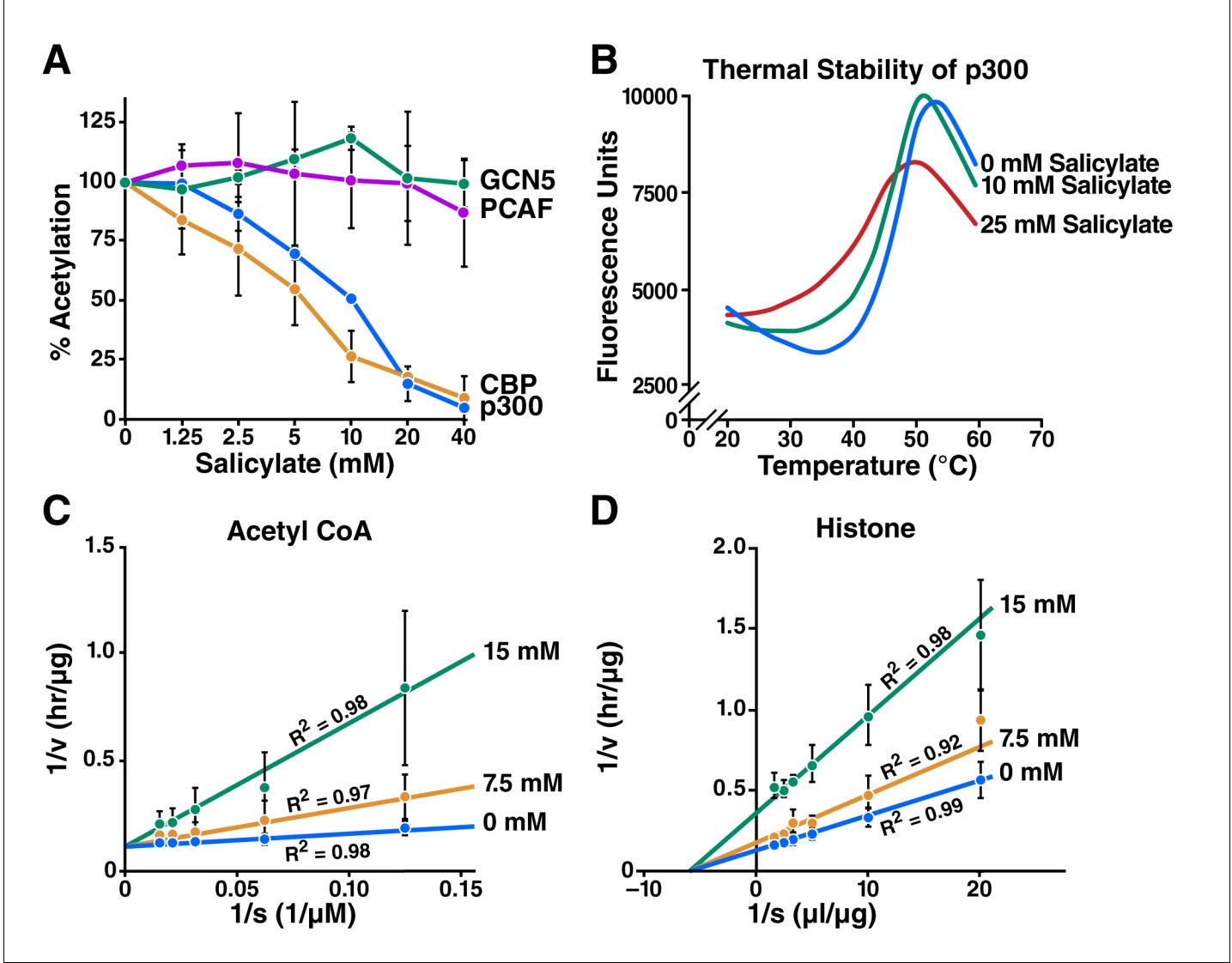

**Figure 1.** Salicylate inhibits CBP/p300 *in vitro*. (A) Recombinant p300, CBP, GCN5, or PCAF and histones were incubated with [14]C-labeled acetyl-CoA with or without sodium salicylate, separated by SDS-PAGE, analyzed by autoradiography, and quantified with Image J software. Acetylation levels are relative to those in untreated controls. (B) Thermal stability assay for sodium salicylate binding to the p300 HAT domain. Tm, melting temperature. (C) and (D) Lineweaver-Burk plots showing kinetic analysis of p300 acetyltransferase inhibition by sodium salicylate. Histone acetylation was measured with several concentrations of acetyl-CoA (C) or histone (D) in the presence or absence of sodium salicylate.

The following figure supplement is available for figure 1:

**Figure supplement 1.** CoA metabolites of salicylate and diflunisal are more potent inhibitors of p300.

48.6°C, while treatment with 10 and 25 mM salicylate reduced the unfolding temperature to 46.1°C and 40.8°C, respectively (*Figure 1B*). Kinetic analysis of p300 acetyltransferase activity with various concentrations of acetyl-CoA (*Figure 1C*) and histone (*Figure 1D*) substrates revealed that salicylate exhibits direct competitive p300 inhibition against acetyl-CoA and noncompetitive inhibition against histones. Taking this data together, we surmised that salicylate inhibits p300 acetyltransferase activity by directly competing with acetyl-CoA binding near its binding site on CBP and p300.

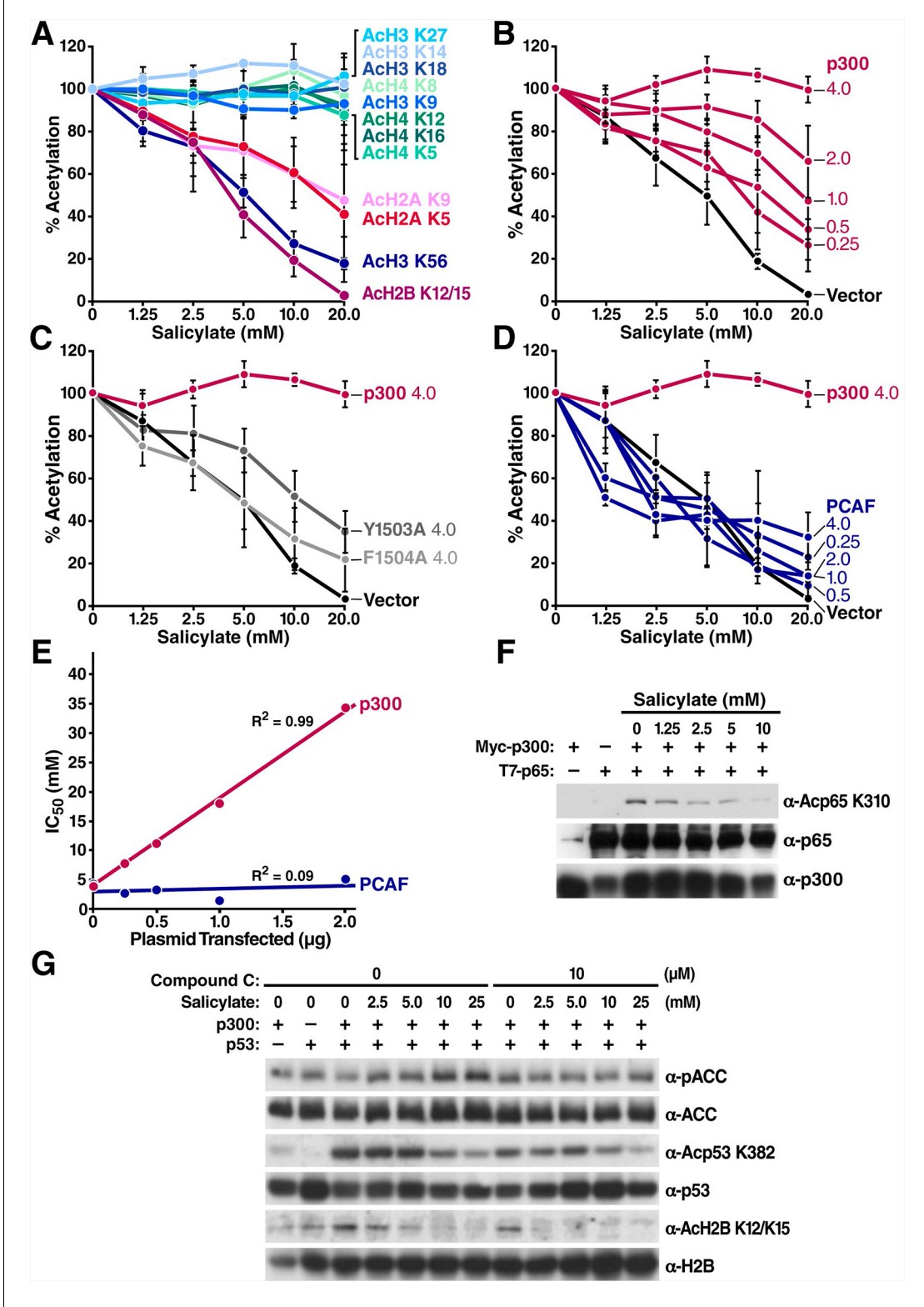

**Figure 2.** Salicylate inhibits specific lysine acetylation of histone and nonhistone proteins independently of AMPK activation. (**A**) Decreased acetylation of specific lysines in histones in the presence of salicylate. HEK293T cells were treated with the indicated concentrations of sodium salicylate for 24 hr. Site-specific histone acetylation was detected by Western blot with specific antisera. Bands were quantified with Image J software. Acetylation was normalized to that of untreated cells and plotted. Representative results are shown in Supplementary *Figure 1*. Experiments are repeated and error bars indicate SEM. (**B–D**) Salicylate-induced hypoacetylation of histone H2B was rescued by overexpression of p300 (**B**) but not by the catalytically

*Figure 2 continued on next page*

*Figure 2 continued*

inactive p300 mutant F1504A (**C**), or PCAF (**D**). HEK293T cells were transfected with increasing amounts of expression vectors for p300 or F1504A or PCAF, treated with sodium salicylate for 24 hr, and analyzed by Western blotting analysis with an antiserum specific for acetyl histone H2B$_{K12/K15}$. Bands were quantified with Image J software. Acetylation was normalized to that of untreated control. Average levels of relative acetylation are plotted and error bars indicate SEM. Representative results are shown in Supplementary *Figure 2—figure supplement 2*. (**E**) IC$_{50}$ values generated from all curves in panel (**B**) and (**D**) were plotted against the amount of plasmid transfected (p300 or PCAF). (**F**), (**G**) HEK293 T cells were transfected with expression vectors for p300 and NF-κB p65 (**F**) or p53 (**G**), treated with salicylate for 24 hr, and analyzed by Western blot with specific antibodies against acetyl NF-κB$_{K310}$ (**F**) or acetyl p53$_{K382}$ and acetyl H2B$_{K12/15}$ (**G**). Compound C (10 µM), a specific AMPK inhibitor, was added to salicylate-treated cells for 24 hr before Western blot (**G**). KR, p65 K310R mutant

The following figure supplements are available for figure 2:

**Figure supplement 1.** Salicylate induces histone deacetylation in HEK293T cells HEK293T cells were treated with sodium salicylate as indicated for 24 hr, immediately lysed in Laemmli buffer, and then subjected to western blot analysis with the indicated antibodies.

**Figure supplement 2.** Salicylate-induced deacetylation of histone H2B can be rescued by overexpression of p300, but not PCAF, in a dose-dependent manner.

## Salicylate inhibits specific lysine acetylation of histone and non-histone proteins independently of AMPK activation

To determine whether salicylate induces histone deacetylation directly in cells, we treated HEK293T cells with various concentrations of salicylate. Western blot analysis with antibodies against various specific acetyl-lysine modifications of histone H2A, H2B, H3, and H4 showed that addition of salicylate correlated with the deacetylation of H2A$_{K5/K9}$, H2B$_{K12/K15}$, and H3$_{K56}$ in a dose-dependent manner (*Figure 2A* and *Figure 2—figure supplement 1*). Other histone residues, including H3$_{K9, K14, K27, K36}$ and H4$_{K5, K8, K12, K16}$, have also been reported to be acetylated by CBP/p300 (*Schiltz et al., 1999*; *Kouzarides, 2007*), but their acetylation state did not change in response to salicylate, possibly as a consequence of redundant activity of other acetyltransferases in the cellular environment (*Kouzarides, 2007*) or opposing effects caused by inhibition of its previously characterized targets. The IC$_{50}$ for salicylate-mediated inhibition of H2B acetylation (4.8 mM) was close to the IC$_{50}$ of CBP measured *in vitro* and to the plasma concentrations of salicylate (1–3 mM) in humans after oral administration (*Goldfine et al., 2010*; *2013*).

To further test the hypothesis that p300 is a relevant target of salicylate *in vivo*, we overexpressed exogenous p300 at different levels and determined whether it suppresses the effect of salicylate on histone H2B acetylation. HEK293T cells were transfected with wild type (WT) p300, catalytically inactive p300 Y1503A and F1504A mutants (*Suzuki et al., 2000*), or PCAF, and the acetylation state of H2B$_{K12/15}$ was assessed after salicylate treatment. Overexpression of WT p300 suppressed the effect of salicylate in a dose-dependent manner and increased H2B$_{K12/K15}$ acetylation (*Figure 2B*), but overexpression of catalytically inactive p300 mutants (*Figure 2C*) and PCAF did not (*Figure 2D* and *Figure 2—figure supplement 2*). Furthermore, the IC$_{50}$ of salicylate strongly correlated with the amount of transfected p300 but not PCAF (*Figure 2E*). These findings support the hypothesis that salicylate-mediated H2B deacetylation is specifically due to inhibition of p300 acetyltransferase activity.

To determine whether salicylate down-regulates the acetylation of non-histone proteins, we overexpressed NF-κB and p53 in 293T cells, treated the cells with salicylate, and assessed acetylation of these proteins with specific antibody against acetyl NF-κB$_{K310}$ and acetyl p53$_{K382}$ (*Figure 2F and G*). Salicylate decreased acetylation of both NF-κB and p53 in a dose-dependent manner. These findings strongly support the hypothesis that p300 acetyltransferase activity is a biologically relevant target for salicylate *in vivo* in cultured cells.

Recently, salicylate was reported to activate AMPK by allosteric binding to its AMP binding site. (*Hawley et al., 2012*) To confirm this finding, we treated HEK293T cells with various doses of salicylate. The levels of a phosphorylated form of acetyl-CoA carboxylase (ACC), an established AMPK target, increased in a dose-dependent manner (*Figure 2G*). Compound C, an AMPK inhibitor, suppressed p-ACC accumulation in response to salicylate, but did not inhibit deacetylation of

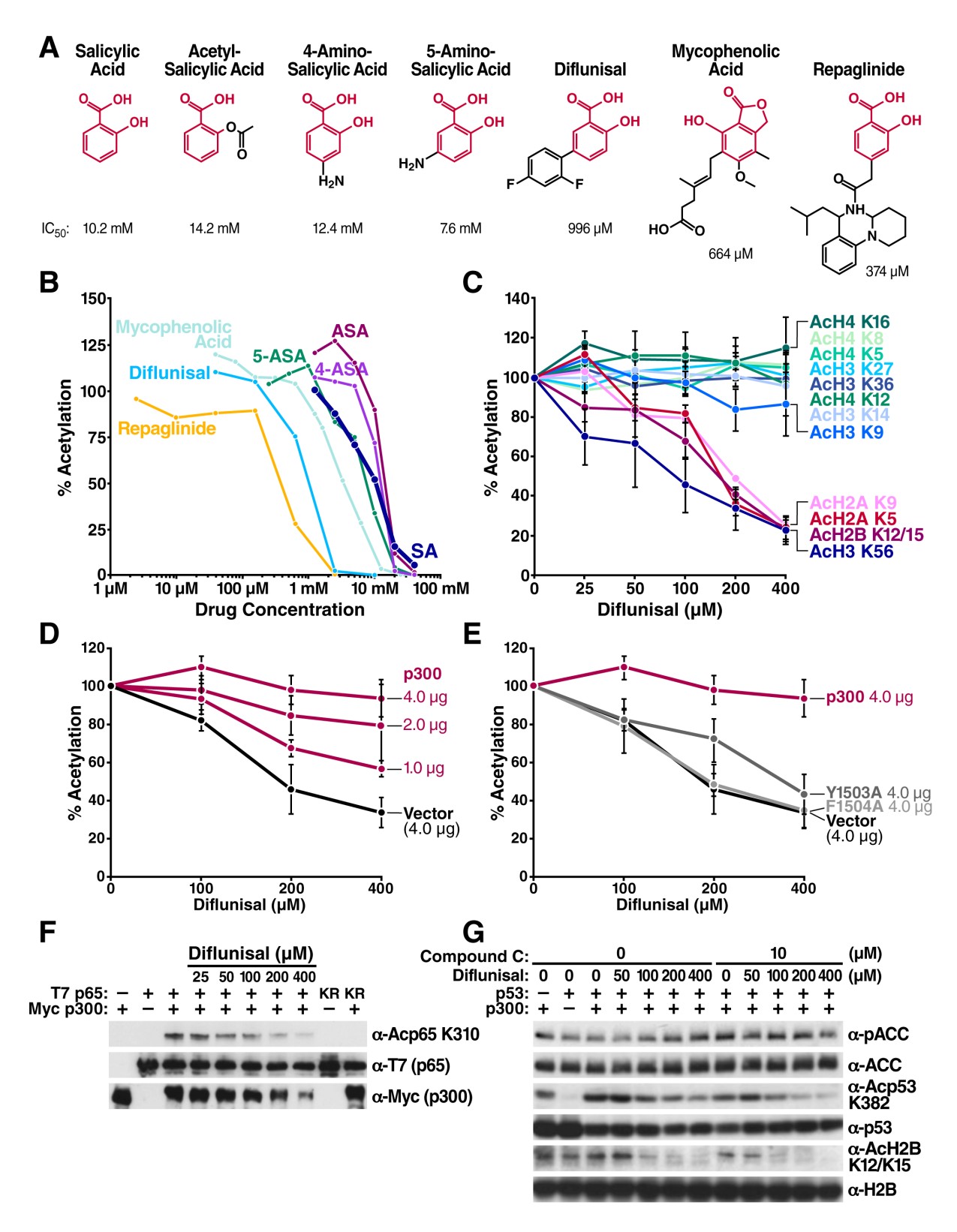

**Figure 3.** Structural homology search identifies diflunisal as a potent p300 inhibitor. (**A**) FDA-approved drugs that contain a structure similar to that of salicylate are shown in red. Numbers below the structures are IC$_{50}$ of each drug, measured by *in vitro* p300 HAT assays. (**B**) Relative HAT activities are

*Figure 3 continued*

plotted by *in vitro* HAT assays using recombinant p300 and histones with increasing amount of various FDA approved drugs. Acetylation levels are relative to those in untreated controls. (C) Relative levels of histone acetylation in response to diflunisal. HEK293T cells were treated with various amount of diflunisal, followed by Western blotting with specific acetyl histone antibodies, as indicated. Bands were quantified with Image J software and plotted. Experiments are repeated two to five times. Error bars indicate SEM. Representative results are shown in *Figure 2—figure supplement 1*. (D) and (E) Diflunisal-induced hypoacetylation of histone H2B was rescued by overexpression of p300 (D) but not by the catalytically inactive p300 mutants Y1503A and F1504A (E) (F) Diflunisal inhibits acetylation of NF-κB p65 (G) and p53 (G) independently of slight AMPK activation (G).

The following figure supplements are available for figure 3:

**Figure supplement 1.** Diflunisal induces histone deacetylation in HEK293T cells HEK293T cells were treated with diflunisal as indicated for 24 hr, immediately lysed in Laemmli buffer, and then subjected to western blot analysis with the indicated antibodies.

**Figure supplement 2.** Diflunisal-induced deacetylation of p300 is rescued by overexpression of p300 in a dose-dependent manner, but not inactive p300 mutants.

acetylated $H2B_{K12/K15}$ or acetyl-$p53_{K382}$. This experiment demonstrates that salicylate-mediated protein deacetylation is not dependent on AMPK activation and activity (*Figure 2G*).

## Structural homology search identifies diflunisal as a potent p300 inhibitor

Next, we tested a series of other drugs that contain a salicylic acid moiety. A substructural homology search of the DrugBank database (www.drugbank.ca) (*Wishart et al., 2006*; *2008*) identified five additional FDA-approved drugs: 4-aminosalicylic acid, 5-aminosalicylic acid, diflunisal, mycophenolic acid, and repaglinide that contain salicylic acid. We tested their ability to inhibit CBP/p300 acetyltransferase activity in *in vitro* HAT assays. All five drugs inhibited p300 with different $IC_{50}$ (*Figure 3A and B*). Three of the drugs inhibited p300 more potently than salicylate: the antidiabetes drug repaglinide ($IC_{50}$ = 374 μM), the immunosuppressant mycophenolic acid ($IC_{50}$ = 664 μM), and diflunisal, an older nonsteroidal anti-inflammatory drug ($IC_{50}$ = 996 μM) (*Figure 3, A and B*).

Since repaglinide induces insulin secretion at nanomolar concentration, it is likely that its ability to inhibit p300 with an $IC_{50}$ of 374 μM is irrelevant to its antidiabetic activity. We therefore selected diflunisal for further analysis. Diflunisal induced deacetylation of specific histone residues, $H2A_{K5, K9}$, $H2B_{K12/K15}$, $H3_{K56}$ (*Figure 3C* and *Figure 3—figure supplement 1*), a pattern of histone acetylation similar to that induced by salicylate. Importantly, the $IC_{50}$ for $H2B_{K12/K15}$ inhibition in cells was 160 μM, which is within the range of plasma concentrations of diflunisal (150–350 μM) after daily oral administration (*Nuernberg et al., 1991*; *Mano et al., 2006*). Overexpression of WT p300 suppressed the effect of diflunisal in a dose-dependent manner and increased $H2B_{K12/K15}$ acetylation (*Figure 3D*, *Figure 3—figure supplement 2*), but overexpression of catalytically inactive p300 mutants did not (*Figure 3E*, *Figure 3—figure supplement 2*). Diflunisal also suppressed acetylation of the nonhistone proteins NF-κB $p65_{K310}$ (*Figure 3F*) and $p53_{K382}$ (*Figure 3G*). Here also, Compound C, an AMPK inhibitor, suppressed p-ACC accumulation in response to diflunisal, but did not inhibit deacetylation of acetylated $H2B_{K12/K15}$ or acetyl-$p53_{K382}$, indicating that AMPK is not necessary for these effects (*Figure 3G*). These findings support the model that diflunisal also targets p300 acetyltransferase activity independently of AMPK.

## Salicylate and diflunisal decrease acetylation of AML1-ETO$_{K43/K24}$ and block the growth of t(8;21) leukemia cells by inducing apoptosis

Previously, we reported that the leukemogenicity of the AML1-ETO fusion protein, generated by a t(8;21) translocation in acute myelogenous leukemia, is regulated by p300-mediated acetylation of lysine 43 of the fusion protein (*Wang et al., 2011*). To investigate a potential application of our newly characterized salicylate- and diflunisal-mediated inhibition of CBP/p300 activity, we tested the effects of various doses of salicylate and diflunisal on two AML1-ETO expressing cancer cell lines (human Kasumi-1 cells and a mouse AE9a-driven AML cell line that we generated). In support of our model, K24 and K43 acetylation of AML1-ETO were decreased in a dose-dependent manner by salicylate (*Figure 4A*, left) and by diflunisal (*Figure 4A*, right). Salicylate inhibited cell proliferation at

concentrations as low as 1 mM (*Figure 4B*). This growth inhibition was caused in part by increased apoptosis, as shown by annexin V/7AAD double staining (*Figure 4C*). Diflunisal also increased apoptosis in a dose-dependent manner (*Figure 4D*). Additional measurements of nuclear DNA distribution showed a dose-dependent increase of the sub-G1 cell fraction, highly suggestive of apoptotic fragmentation by salicylate (*Figure 4E*) and diflunisal (*Figure 4F*). We also noted an increased fraction of G1 cells and a decreased fraction of S and G2/M cells after salicylate treatment, consistent with reports that CBP/p300 is required for the G1/S transition (data not shown) (*Ait-Si-Ali et al., 2000*; *Iyer et al., 2007*). Salicylate did not affect the surface expression of differentiation-related antigens (CD11b and CD34 in Kasumi-1, Mac-1 and C-kit in AE9a) (data not shown), in accordance with our previous finding that acetylation of AML1-ETO is required for self-renewal and leukemogenesis but not for its ability to block cell differentiation (*Wang et al., 2011*). To further test whether p300 is the relevant target of diflunisal in Kasumi-1 cells, we transduced lentiviral expression vectors for p300 or empty control into Kasumi-1 cells (*Figure 4G*). Cells transduced with the empty vector showed inhibition of growth by diflunisal, similar to untransduced cells (*Figure 4H*). In contrast, p300-transduced cells were significantly more resistant to diflunisal (*Figure 4H and I*), and exhibit less apoptotis measured by annexin V positive cells (*Figure 4J*) and sub-G1 fraction (*Figure 4K*). These results support the model that diflunisal kills Kasumi-1 cells by apoptosis due to p300 inhibition.

## Salicylate and diflunisal inhibit AML1-ETO leukemia cell growth in mice

Finally, to examine whether diflunisal inhibits leukemia development *in vivo*, we inoculated SCID mice with Kasumi-1 cells. Starting 3 weeks after inoculation, the mice were treated daily with diflunisal (50 or 100 mg/kg orally) or vehicle. Diflunisal reduced tumor volumes in a dose-dependent manner (*Figure 5A*) and had minimal effects on body weight (*Figure 5B*). After 3 weeks of treatment, the tumors were significantly smaller in diflunisal-treated mice than in vehicle-treated controls (*Figure 5C*), and most of the tumors had disappeared in mice treated with the higher dose of diflunisal (*Figure 5D*).

## Discussion

This study shows that salicylate inhibits CBP/p300 acetyltransferase activity by directly competing with acetyl-CoA, and it down-regulates the specific acetylation of histones and non-histone proteins in cells. We also found that diflunisal, an FDA-approved drug containing a salicylic acid substructure, inhibited CBP/p300 more potently than salicylate. Both drugs inhibited p300-dependent AML-ETO leukemic cell growth *in vitro* and *in vivo*. Thus, diflunisal and salicylate have promise as a oral therapy for patients with acute myelogenous leukemia associated with a t(8;21) translocation, an exciting potential application of our observations..

In plants, from where it was originally isolated, salicylate acts as an immune signal to induce systemic acquired resistance. It specifically activates the transcription cofactor NPR1 (nonexpressor of PR genes 1) by binding to its paralogs NPR3 and NPR4 (*Fu et al., 2012*). Since plants contain an ortholog of p300/CBP (*Bordoli et al., 2001*), some of its activities in plants could also be mediated by inhibition of the plant ortholog of p300 or CBP.

In animals, salicylate is an extensively studied small compound widely used as an anti-inflammatory drug. Many mechanisms of action have been proposed for the anti-inflammatory effects of salicylate in mammalian cells, including weak inhibition of cyclooxygenase (*Warner et al., 1999*), inhibition of IKKβ inhibition (*Yin et al., 1998*) and topoisomerase II (*Cox et al., 2011*), modulation of NF-κB (*Kopp and Ghosh, 1994*), and activation of AMPK (*Hawley et al., 2012*). The simple structure of salicylate might enable it to interact with different affinities with many cellular proteins, which could explain its pleiotropic effects. While we cannot completely rule out the possibility that the effects of salicylate on acetylation may derive in part from off-target effects, in addition to the direct interaction with CBP/p300 reported here, our study demonstrate a direct inhibition of p300 and CBP by salicylate, diflunisal and their metabolites (see discussion below).

We observed that the acetylation of both histones and non-histone proteins (NF-κB) is suppressed in cells treated with either salicylate or diflunisal. We identified histone AcH3K56 as the most sensitive histone acetyl mark to inhibition by both drugs. This results is highly consistent with the literature showing that CBP (also known as Nejire) in flies and CBP and p300 in humans acetylate

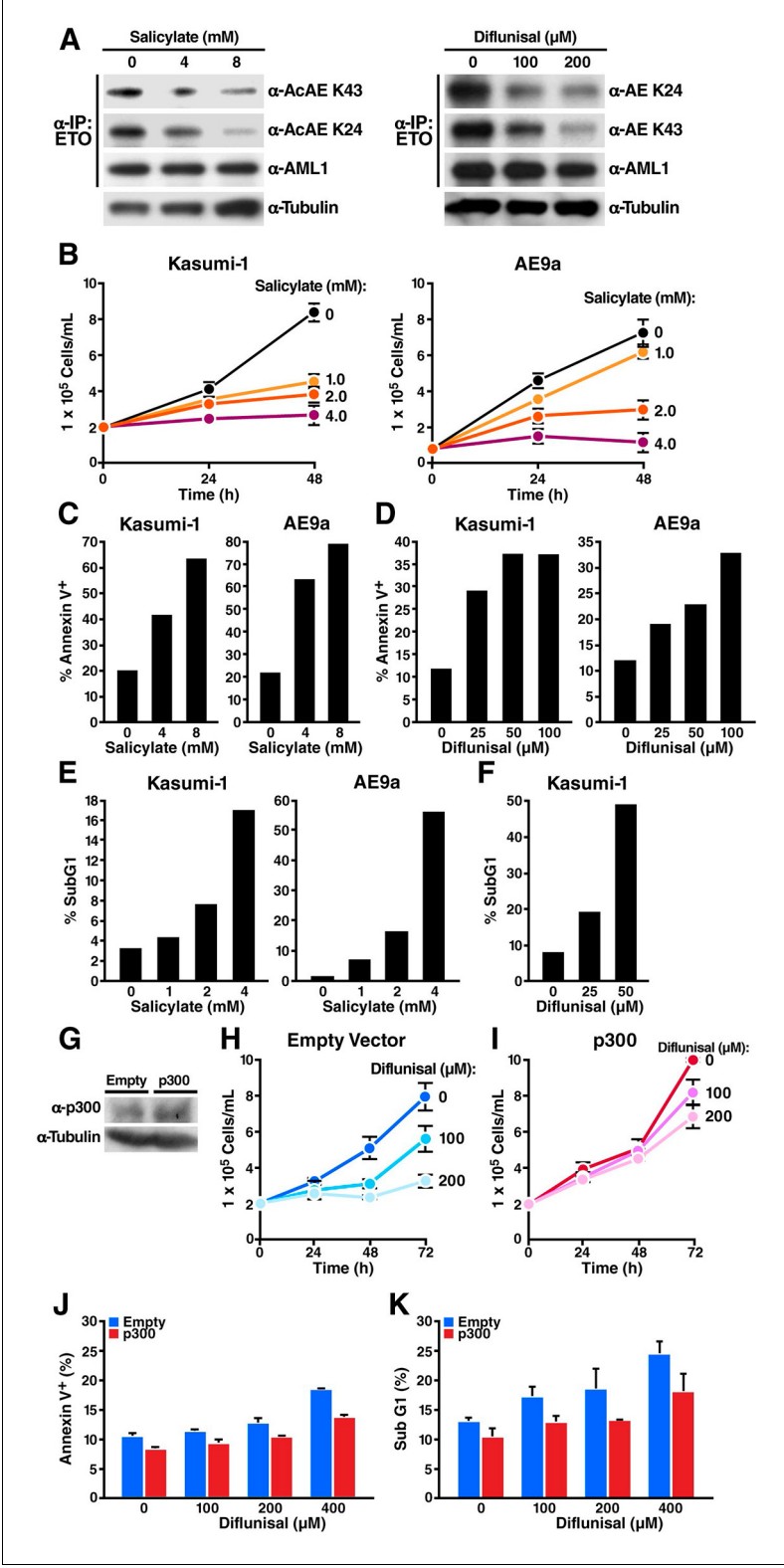

**Figure 4.** Sodium salicylate and diflunisal decrease acetylation of AML1-ETO$_{K43/K24}$ and block the growth of t(8;21) leukemia cells by inducing apoptosis. (**A**) Kasumi-1 cells expressing the AML1-ETO fusion protein were treated with sodium salicylate (4 or 8 mM, *left*) or diflunisal (100 or 200 μM, *right*) for 24 hr, followed by immunoprecipitation of AML1-ETO and analysis by Western blotting with an anti-acetyl lysine antiserum. (**B**) Kasumi-1 and AE9a cells treated or not with salicylate were counted by trypan-blue exclusion under light
*Figure 4 continued on next page*

*Figure 4 continued*

microscopy. (**C**) and (**D**) Annexin-V/7AAD staining of Kasumi-1 and AE9a cells after 24 hr of treatment with salicylate (**C**) or diflunisal (**D**). (**E**) and (**F**) Kasumi-1 and AE9a cells treated or not with salicylate (**E**) or diflunisal (**F**) were collected, and DNA content was measured by propidium iodide staining after overnight fixation in 75% ethanol. The percentage of sub-G1 cells is shown. (**G**) p300 overexpression in the p300 lentiviral tranduced cells were confirmed by Western Blotting. (**H**) and (**I**) Kasumi-1 cells were transduced with p300 or empty lentiviral vector and treated with or not diflunisal. Cells were counted by trypan-blue exclusion under light microscopy. (**J**) and (**K**) Annexin-V/7AAD staining (**J**) and subG1 population analyzed by PI staining (**K**) of p300 transduced Kasumi-1 cells after 6 hr of treatment of diflunisal.

---

H3K56 (*Das et al., 2009*). We also note that the pattern of histone marks inhibition are remarkably similar between the two drugs, but at different concentrations in agreement with their relative abilities to inhibit p300/CBP *in vitro*. In terms of what is observed for the other histone aceylated sites, the situation is more complex. Indeed, many histone modifications are regulated by multiple HAT enzymes. For example, acetyl H4K5 is regulated by HAT1, CBP, p300, Tip60, HB01 whereas acetyl H3 K14 is regulated by CBP, p300, PCAF, gcn5, ScSAS3 (*Kouzarides, 2007*). We therefore interpret the observed lack of inhibition of some histone H3 or H4 acetylation sites by salicylate or diflunisal to reflect the compensating activities or other histone acetyltransferases that target the same sites.

We also observed inhibition of NF-κB acetylation by both salicylate and diflunisal (*Figures 2D* and *3D*). The NF-κB subunit RelA is acetylated on lysines 218, 221, and 310 and these modifications are required for full NF-κB activation (*Chen et al., 2001*; *2002*). Therefore, salicylate's NF-κB inhibitory effect can be at least partly explained by p300 inhibition. AMPK is reported to inhibit p300 acetyltransferase activity by phosphorylating p300 at serine 89 (*Zhang et al., 2011*), suggesting that AMPK activation by allosteric salicylate binding might inhibit p300 indirectly. However, our findings clearly showed that salicylate does not inhibit p300 by activating AMPK since the p300 inhibition is insensitive to an AMPK inhibitor (*Figure 2G*).

Overall, our findings suggest that many of the pleiotropic effects of salicylate on different cell types and in diseases, including leukemia, are mediated by specific inhibition of CBP/p300 acetyltransferase activity, leading to deacetylation of histones and non-histone proteins.

Interestingly, the $IC_{50}$ for p300 inhibition by both salicylic acid and diflunisal was significantly lower in HEK293T cells (*Figure 2A*) than in HAT assays *in vitro* (*Figure 1A*). A number of mechanisms discussed below could account for these differences.

First, AMPK activation in cells might enhance the p300 inhibitory effect of salicylate in cells (*Zhang et al., 2011*).

Second, we have also observed that short-term treatment of cells with salicylate or diflunisal is associated with a decreased in the expression p300 (see *Figure 2F* and *3D*). This phenomenon is accentuated when cells are treated longer with salicylate or diflunisal (data not shown). Other molecules have been shown to induce p300 degradation through the activation of different signaling transduction cascades (*Chen and Li, 2011*) and the autoacetylation of p300 is important for its enzymatic activity (*Thompson et al., 2004*). While we have not further explored this interesting observation here, we could envisage a mechanism by which inhibition of p300 autoacetylation would both contribute to the inactivation of the enzyme but also to a change in its stability.

Third, metabolism of salicylic acid and diflunisal may also contribute to increased cellular potency *in vivo*. Indeed, we have found that salicyl-CoA, a known major intermediate of salicylate metabolism (*Knights et al., 2007*), inhibits CBP/p300 with 28-fold increased potency in comparison with salicylate: $IC_{50}$=220 μM for salicyl-CoA vs 6.12 mM for salicylate (*Figure 1—figure supplement 1*). A similar 52-fold increase in potency is observed with diflunisal-CoA in comparison to diflunisal: $IC_{50}$=20 μM for diflunisal-CoA vs 1.05 mM for diflunisal (*Figure 1—figure supplement 1*). Although further investigation will be required to understand the relative contribution of both salicylic acid, diflunisal and their metabolites to the novel *in vivo* effects of salicylate reported here, these observations provide a potential mechanistic basis for the potent cell-based effects of these compounds.

Fourth, it should be noted that a fragment of p300 consisting of its HAT domain is used in our *in vitro* experiments and in all other published studies using recombinant p300 protein. It is possible that the sensitivity of this subdomain to salicylate inhibition might be significantly different from the full-length protein present in cells.

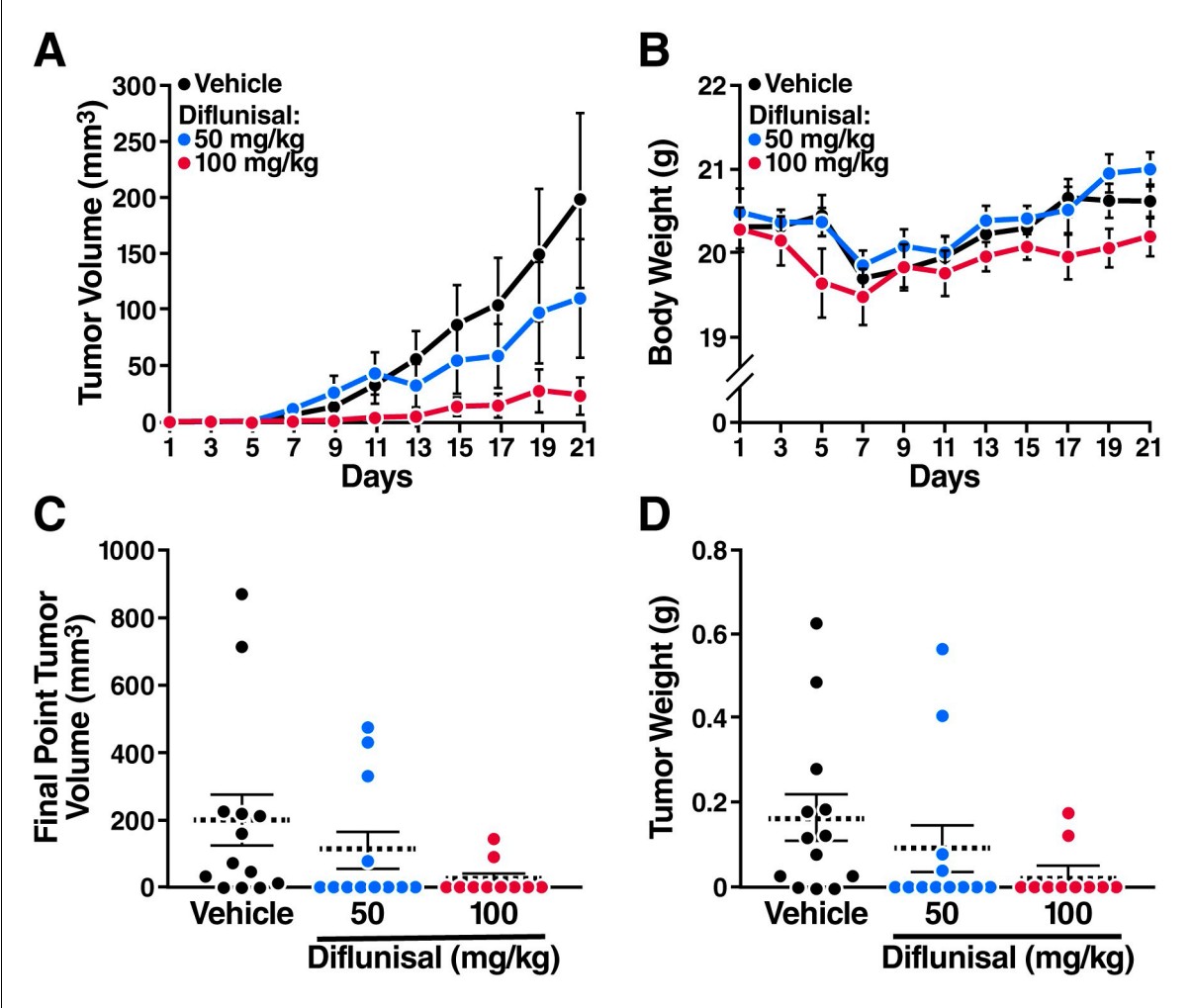

**Figure 5.** Sodium salicylate and diflunisal inhibit the growth of AML1-ETO leukemia cells in SCID mice. The mice were inoculated with Kasumi-1 cells ($3\times10^7$) and, starting 3 weeks later, were treated daily with oral diflunisal (50 or 100 mg/kg) or vehicle for 3 weeks. (A) and (B) Plots of tumor volume (A) and body weight (B). (C) and (D) Tumor size (C) and body weight (D) at sacrifice after 3 weeks of treatment.

Most of the non steroidal anti-inflammatory drugs (NSAIDs) inhibit both COX-1 and COX-2, although they vary in their relative potencies against the two isozymes (*Patrignani et al., 1997*). However, salicylate does not, unlike its acetylated derivative aspirin, inhibit COX-1 and COX-2 activity *in vitro* (*Vane, 1971*) (*Vargaftig, 1978*; *Mitchell et al., 1993*); (*Cromlish and Kennedy, 1996*). However, salicylate has been shown to exert a comparable analgesic and anti-inflammatory action as aspirin. Salicylates have been proposed to exert their pharmacological effects via inhibition of the transcription factor nuclear factor NF-|B and other targets. In these experimental systems, the concentrations used were in the same range as used in the experiments described in our paper (5–20 mM) (*Kopp and Ghosh, 1994*) (*Pierce et al., 1996*; *Oeth and Mackman, 1995*; *Schwenger et al., 1996*; *1997*; *1999*).

Further, it should also be noted that salicylate plasma concentrations in patients taking salicylic acid (3–4 g/day) range between 1–3 mM, a concentration at which partial inhibition of p300/CBP is observed. These data are therefore consistent with the proposed model that partial or complete inhibition of p300 by salicylate represents one of its relevant biological targets. It is important to note that a small molecule the size of salicylic acid used at such high concentrations is expected to interact with a number of cellular proteins and that our discovery that salicylic acid targets p300 does not imply that previous targets are not also part of the cellular response to these drugs.

Several other FDA-approved drugs with substructures similar to that of salicylate also directly inhibited p300 acetyltransferase activity (*Figure 3A*). The salicylate substructure is now identified in three HAT inhibitors, including anacardic acid (6-nonadecyl salicylic acid) (*Balasubramanyam et al., 2003*), salicylic acid and diflunisal and might therefore represent an important scaffold for developing new p300 inhibitors.

Recently, p300 has emerged as a potential therapeutic target for respiratory diseases, HIV infection, metabolic diseases, and cancer (*Dekker and Haisma, 2009*). Indeed, our findings show that salicylate and the related compound diflunisal exhibit anti-tumor activity against a specific leukemia carrying a t(8;21) translocation, a tumor previously reported to be dependent on p300 *in vitro* and *in vivo* (*Wang et al., 2011*). We have tested whether other NSAIDs, including acetaminophen and indomethacin, also inhibit p300 acetyltransferase *in vitro*, but did not detect any inhibitory activity (data not shown). Importantly, NSAIDs that lack p300 inhibitory activity failed to inhibit Kasumi-1 cells growth (data not shown).

These results identify a novel epigenetic therapeutic target for salicylate, the epigenetic regulator CBP/p300. Further efforts will focus on unraveling the relative roles of different cellular targets of salicylate, such as CBP/p300, cyclooxygenases, IKKβ, and AMPK. Our results also suggest that salicylate may be useful for treating inflammation, diabetes, neurodegenerative disease, and other pathologies in which CBP/p300 has a critical role.

## Materials and methods

### *In vitro* HAT assay

Recombinant HAT (1 mg), either p300, CBP, PCAF, or GCN5 (Enzo Life Sciences), and 10 µg of histones (Sigma) were incubated with sodium salicylate (Sigma) in reaction buffer (50 mM HEPES, pH 8.0, 10% glycerol, 1 mM) at 30°C for 30 min and then with 0.1 mCi of $^{14}$C acetyl-CoA at 30°C for 60 min. Reactions were stopped by adding 6x sample buffer and analyzed by SDS-PAGE. The gels were dried, and signals were obtained by autoradiography and quantified with Image J software. To quantify acetylated histone levels, we generated a standard curve from signals of lanes loaded with 2.5, 5, and 10 g of $^{14}$C labeled histones.

### Thermal stability assay

A thermal stability assay was used to assess the binding of salicylate to the p300 HAT domain. A p300 HAT domain construct (residues 1279–1666) bearing an inactivating Tyr1467Phe mutation to facilitate purification of homogeneously hypoacetyalted p300 was cloned into a pET-DUET vector with an N-terminal 6-His tag and expressed in BL21 (DE3) *E. coli* cells. Cells were grown at 37°C until they reached an optical density (600 nm) of 0.8, incubated with 0.5 mM IPTG (Isopropyl β-D-1-thiogalactopyranoside) overnight at 18°C to induce protein expression, harvested, and lysed by sonication in lysis buffer (25 mM HEPES, pH 7.5, 500 mM NaCl, and 5 mM ®-mercaptoethanol). The lysate was cleared by centrifugation and applied to a Ni-NTA affinity column. The protein was eluted from the column with increasing concentrations of imidazole in lysis buffer (20–250 mM) and treated overnight with TEV protease to cleave the 6-His tag. The protein was further purified by passage through a HiTrap SP HP ion-exchange column and a size-exclusion Superdex 200 column equilibrated with 25 mM HEPES, pH 7.5, 150 mM NaCl, and 5 mM -mercaptoethanol.

X-ray crystallography showed that the purified p300 HAT domain protein binds to acetyl-CoA or CoA (apparently from the bacterial cell) (*Maksimoska et al., 2014*) In thermal stability experiments in a 384-well ABI plate (Applied Biosystems), the p300 HAT domain bound to acetyl-CoA/CoA was incubated with increasing concentrations of sodium salicylate for 30 min. The final concentration of p300 was 2 µM in reaction buffer (0.1 M HEPES, pH 7.5, 150 mM NaCl, and 5 mM -mercaptoethanol). Then, 4 µl of a 1:200 dilution of stock SYPRO orange dye (Invitrogen) in reaction buffer was added to achieve a total reaction volume of 20 µl. Thermal melt curves were obtained by heating the protein from 20–95°C and monitoring fluorescence at 590 nm with a 7900HT Fast Real Time PCR System (Applied Biosystems). All curves were obtained in triplicate and averaged.

## Evaluation of NSAID-CoA metabolites as p300 inhibitors

Salicyl-CoA and diflunisal-CoA were synthesized from their parent carboxylic acids and HPLC purified according to previously reported methods (*Padmakumar et al., 1997*). The purity of each acyl-CoA was confirmed by analytical HPLC prior immediately prior to utilization (*Montgomery et al., 2014*; *Fanslau et al., 2010*). P300 inhibition assays were performed using direct microfluidic mobility shift analysis as previously described. Briefly, p300 reaction mixture (50 mM HEPES, pH 7.5, 50 mM NaCl, 2 mM EDTA, 2 mM DTT, 0.05% Triton-X-100, 50 nM p300, 2 μM FITC-histone H4 peptide) was plated in 384-well plates and allowed to equilibrate at room temperature for 10 min. Reactions were initiated by addition of acetyl-CoA (final concentration = 5 μM), bringing the final assay volume to 30 μL. Assays were quenched after 10 min (<15% product accumulation) by addition of 5 μL of 0.5 M neutral hydroxylamine and transferred to a Perkin-Elmer Lab-Chip EZ-Reader instrument for analysis. Separation conditions were: downstream voltage of −500 V, upstream voltage of −2500 V, and a pressure of −1.5 psi. Percent conversion was calculated by ratiometric measurement of substrate/product peak heights. Percent activity represents the percent conversion of KAT reactions treated with inhibitors relative to untreated control KAT reactions, and corrected for nonenzymatic acetylation. Dose-response analysis of p300 inhibition was performed in triplicate and analyzed by nonlinear least-squares regression fit to $Y = 100/(1 + 10\wedge(Log\ IC50 - X)*H)$, where H = Hill slope (variable). IC50 values represent the concentration that inhibits 50% of KAT activity. Calculations were performed using Prism 6 (GraphPad) software.

## Cell culture

HEK293T cells (ATCC) were maintained in DMEM supplemented with 10% FCS. The viability of Kasumi-1 cells and AE9a mouse leukemia cells was assessed in triplicate by trypan blue exclusion. The Kasumi cell line was isolated and characterized by one of the coauthors (S. Nimer) (*Becker et al., 2008*). All cell lines were tested annually for mycoplasma contamination. Only negative mycoplasma cultures were used during the conduct of these experiments.

## Plasmids

pCi-p300 and pCi-PCAF are described elsewhere (*Boyes et al., 1998*). pcDNA3/myc-p300 and pcDNA3/T7-p65 were described previously (*Chen et al., 2001*). pCi-p300 Y1503A, F1504A and pcDNA3/T7-p65 K310R were constructed by using the QuickChange site-directed mutagenesis kit (Promega). Lentiviral plasmids, pCSII-CMV-MCS, pCAG-HIVgp, pCMV-VSV-G-RSV-Rev are kindly provided by H Miyoshi, RIKEN BioResource Center, Tsukuba, Japan (*Bai et al., 2003*). p300 CDS was cloned into XhoI/NotI site of pCSII-CMV-MCS.

## Western blot and antibodies

HEK293T cells were treated with sodium salicylate for 24 hr and lysed in lysis buffer (25 mM Tris, pH 6.8, 2% SDS, and 8% glycerol). For Western blot analysis, we used antibodies against acetyl histone H2A$_{K5}$ (ab1764, Abcam), acetyl histone H2A$_{K9}$ (ab47816, Abcam), acetyl histone H2B$_{K12/K15}$ (ab1759, Abcam), acetyl histone H3$_{K9}$ (06–942, Millipore), acetyl histone H3$_{K14}$ (12–359, Millipore), acetyl histone H3$_{K27}$ (07–360, Millipore), acetyl histone H3$_{K36}$ (07–540, Millipore), acetyl histone H3$_{K56}$ (2134–1, Epitomics), acetyl histone H4$_{K5}$ (06-759-MN, Millipore), acetyl histone H4$_{K8}$ (06-760-MN, Millipore), acetyl histone H4$_{K12}$ (6-761-MN, Millipore), acetyl histone H4$_{K16}$ (06-762-MN, Millipore), histone H2A (07–146, Millipore), histone H2B (ab1790, Abcam), histone H3 (07–690, Millipore), histone H4 (07–108, Millipore), p300 (ab3164, Abcam), PCAF (ab96510, Abcam), tubulin (T6074, Sigma), acetyl NF-κB p65$_{K310}$ (3045, Cell Signaling), NF-κB p65 (sc-372, Santa Cruz Biotechnology), and acetyl lysine (9441, Cell Signaling).

## Lentiviral transduction

pCSII-CMV-MCS vectors and packaging plasmids were transfected to HEK293T cells, supernatant were collected and ultracentrifuged 48 hr after transfection. Same amount as p24 levels of lentiviruses contain empty or p300 expression vectors were transduced to Kasumi-1 cells.

## Immunoprecipitation assay

Kasumi-1 cells were treated with sodium salicylate or diflunisal for 24 hr and lysed in RIPA buffer. AML1-ETO protein in the lysate was immunoprecipitated with an anti-ETO antibody (Santa Cruz Biotechnology). Antibodies against AML1 and acetylated AML1-ETO K24/K43 (generated in the Nimer lab) were used for Western blotting.

## Flow cytometry

Apoptosis was analyzed with an Annexin V-APC/7AAD Apoptosis kit (Becton-Dickinson) according to manufacturer's instructions. To assess the distribution of nuclear DNA content in the cell-cycle analysis, cells were collected, washed in PBS, fixed overnight in 75% ethanol at –20°C, treated with 1% RNase A for at least 15 min at 37°C, and stained with 50 mg/ml propidium iodide. To monitor CD34 expression, the cells were stained with an allophycocyanin-conjugated anti-CD34 antibody (Becton Dickinson). To monitor CD11b expression, the cells were stained with a phycoerythrin-conjugated anti-CD11b antibody (Beckman-Coulter). To monitor C-kit and Mac-1 expression, the cells were stained with allophycocyanin-conjugated anti-C-kit and phycoerythrin-conjugated anti-Mac-1 antibodies (Becton-Dickinson). Cells were sorted with a Becton-Dickinson FACSCalibur, and the data were analyzed with FlowJo software.

## Xenograft model

Severe combined immunodeficiency (SCID) mice were injected with 30 million Kasumi-1 cells in 100 μl of PBS and 100 μl of Matrigel. Three weeks after inoculation, when tumors can be detected, mice were treated daily with oral diflunisal (50 or 100 μg/kg) or vehicle. Tumor volume and body weight were recorded every 2 days. After 3 weeks of treatment, the mice were killed and tumor size and body weight were recorded. All mice were maintained according to National Institutes of Health guidelines and all animal use protocols were approved by the Institutional Animal Care and Use Committee.

## Acknowledgements

We thank John Carroll and Teresa Roberts for graphics and Veronica Fonseca for help with manuscript preparation. This work is supported by funds from the Gladstone Institutes. KS is supported by a grant from the National Institutes of Health (NIH), the University of California San Francisco (UCSF)-Gladstone Institute of Virology & Immunology Center for AIDS Research, P30 AI27763, and the University of California, Berkeley Fogarty International AIDS Training Program (AITRP). LW is supported by an Institutional Research Grant (IRG-98-277-13) from the American Cancer Society. JCN is supported by fellowships from Larry L Hillblom Foundation, the John A Hartford Foundation, and the UCSF Geriatric Research Training Program (NIA 5T32AG000212-20). MO is supported by a grant from the NIH (AI083139-02).

## Additional information

### Funding

| Funder | Grant reference number | Author |
| --- | --- | --- |
| National Institutes of Health | AI083139-02 | Melanie Ott<br>Kotaro Shirakawa |
| UCSF Gladstone Institute of Virology & Immunology Center for AIDS research | NIA 5T32AG000212-20 | Kotaro Shirakawa |
| University of California Berkeley Fogarty International AIDS Training Program | | Kotaro Shirakawa |
| American Cancer Society | | Lan Wang |
| Larry L. Hillblom Foundation | | John C Newman |
| Jonh A. Hartford Foundation | | John C Newman |

UCSF Geriatric Research
Training Program

John C Newman

The funders had no role in study design, data collection and interpretation, or the decision to submit the work for publication.

### Author contributions
KS, LW, NM, JMa, AWS, HWL, ISL, TS, JCN, SS, MO, RM, JMe, SN, EV, Final approval of the version to be published, Conception and design, Acquisition of data, Analysis and interpretation of data, Drafting or revising the article

### Author ORCIDs
Eric Verdin, http://orcid.org/0000-0003-3703-3183

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
