## [Decision Letter]

Thank you for submitting your study titled, "Salicylate and diflunisal are CBP/p300 inhibitors with anticancer activity" for consideration by *eLife*. I am happy to report that your manuscript has been reviewed by three experts in the field and by a member of the Board of Reviewing Editors (BRE) who are unanimously excited about the findings reported in this manuscript. The evaluation was overseen by Randy Schekman as the Senior Editor.

Furthermore, the reviewers have discussed the reviews with one another, and the BRE member and we hope that you will be able to address their comments included below and to submit the revised version within two months. We have also included at the bottom of this letter the entire reviews provided by the reviewers. Please let us know if you have any questions in this regard.

We urge you to consider the following additional efforts:

1) Perform further studies on the analysis of diflunisal to help explain the differences observed in its IC_50_ in vivo and in vitro as this information would make the paper much stronger.

2) Perform further studies to rule out off-target effects perhaps by characterizing one or two of the commonly known targets of diflunisal.

Reviewer #1:

In this manuscript, Shirakawa and coworkers characterized salicylate and diflunisal as novel CBP/p300 inhibitors. This finding is surprising but also important for several reasons. First, salicylate is widely known for its anti-inflammatory property by targeting mainly COX1/2 and other proteins. The finding that this molecule also acts on CBP/p300 adds another functionally important target to appreciate the role of salicylate. Second, the Nimer lab previously showed the importance of CBP/p300 for the survival of ETO AML and herein demonstrated that salicylate and diflunisal are effective against ETO AML at the dose relevant to depletion of several acetylation marks. The therapeutic implication of the two compounds is important given their ready use, although the two compounds may hit multiple other targets at the present high doses.

The main concerns of the reviewer are:

1) Although the role of salicylate against CBP/p300's acetylation activity was rigorously examined by the key rescue experiment shown in Figure 2, diflunisal hasn't been characterized to the same degree. It seems that the authors solely based on the observation that diflunisal is structurally similar to salicylate and then concluded that the two compounds act via the same modes of mechanism. The experiments in Figure 1 and in particular 2B, C and D should be performed for diflunisal as well.

2) For the efficiency of the two compounds against ETO AML, it is not clear whether the effect is achieved by targeting CBP/p300 alone or other targets alone or multiple targets in a combined manner. If the latter are the case, the flow of the work will be disrupted and the impact of the work will be significantly dampened (the potential target(s) of the therapeutic outcome of the two compounds were not CBP/p300). The reviewer strongly feels that the rescue experiments with wild-type and catalytically dead p300 should be carried out at a cellular level in parallel with the experiments in Figure 4 (required) and even better for the animal experiments in Figure 5.

Reviewer #2:

Shirakawa et al. assess salicylate derivatives as potential p300/CBP HAT inhibitors based on the fact that other recently developed HAT inhibitors contain a salicylic acid moiety. They first show that salicylate inhibits p300/CBP Hat activity in vitro with an IC_50_ in the 10 mM range. They show that they can inhibit some, but not other histone modifications that are known to be a result of p300/CBP in cells. They go on to identify the small molecule diflunisal as a Hat inhibitor and show that it can inhibit acetylation of AML1-ETO and growth in mice. The data are of interest but there are significant issues regarding specificity of mechanism.

1) The IC_50_ of 10mM for salicylate seems high. Can the authors put this into some context as to the concentrations at which salicylate works against other targets?

2) The fact that salicylate inhibits acetylation of specific histone residues is interesting, but the fact that not all p300/CBP dependent modifications are inhibited is concerning in terms of mechanism. The authors explain this by potential redundant enzymes. Anything that can be done here to help explain this result would be very helpful.

3) A significant concern in Figure 3 is that the in vitro IC_50_ for diflunisal is that the in vitro IC_50_ is 1mM and the cellular half-maximal inhibition is 100µM. Can this be explained? It seems strange that the IC_50_ in cells would be less than the IC_50_ in vitro.

4) The data on inhibition of AML1-ETO cell growth in vitro and in vivo is of interest, however there still remains a specificity issue. The authors need to test growth inhibition on other cell lines without AML1_ETO rearrangement or show that indeed inhibition of AML1-ETO acetylation is the reason for the antiproliferative effect. Also, there needs to be assessment of inhibition of acetylation in the leukemia cells in vivo.

Reviewer #3:

The IC_50_ of salicylate for p300/CBP seems to be too high to be specific for HAT/KATs. The authors should show the specificity, using in vivo acetylation of nonhistone substrate, such as p53 acetylation by p300/CBP and PCAF (specific commercial antibodies are available).

Is it really inhibiting KAT activity in the animal system? The authors may consider to feed the animal (mice) and do immunohistochemistry (IHC) analysis of liver tissue.

Inhibition of cancer progress has been claimed by the authors. The data is quite convincing. Is it possible to show that it is due the inhibition of p300/CBP KAT activity by salicylate?

[Editors' note: further revisions were requested prior to acceptance, as described below.]

Thank you for resubmitting your work entitled, "Salicylate and diflunisal are CBP/p300 inhibitors with anticancer activity," for further consideration at *eLife*. Your revised article has been favorably evaluated by Randy Schekman as the Senior Editor, a Reviewing editor, and three reviewers. Please note that the reviewers have requested modifications in the text that need to be implemented before the manuscript can be accepted. We have included the reviewers' comments below for you and look forward to receiving your revised manuscript and the rebuttal letter fully addressing the reviewers' comments.

Reviewer #1:

The revised paper addressed the most of the previous concerns. In particular, the new model of the formation of "salicyl-CoA" and "diflunisal-CoA", if they are indeed the cases, may explain many conflicting data in vitro. However, for such situations, the in vitro biochemical data may not be relevant anymore to evaluate these compounds because they will act differently upon the formation of CoA-adduct. In contrast, the weak interaction of salicyl or diflunisal only argues that HATs contain binding pockets to accommodate these moieties. The reviewer strongly suggests revising the wording of "salicyl or diflunisal as inhibitors of HATs)" into "HATs contain the binding pockets of salicyl or diflunisal to engage the corresponding interaction in vitro and in cellular contexts". By this way, it leaves the possibility that salicyl or diflunisal may be subject to the formation of CoA-adduct and thus act as pro-drugs inside cells. The paper can even be further strengthened by showing the formation of "salicyl-CoA" and "diflunisal-CoA" in a cellular setting.

Reviewer #2:

The manuscript is improved as a result of the new experiments provided. However, while the authors respond to all of the concerns by the reviewers, there is no evidence that these discussions made it into the manuscript. It would be helpful if the authors added some discussion in the manuscript that focus on the reviewers’ questions given it is likely readers will have the same questions.

Also, it is not clear why the authors did not expand their assessment of cell lines beyond Kasumi/AML1-ETO. This is not difficult and thus makes one wonder if the compound it toxic to many cell lines.

Reviewer #3:

The authors have addressed most of reviewers’ concerns. I am satisfied with the revised version of the manuscript. It can be accepted for publication.

---

## [Author Response]

*We urge you to consider the following additional efforts:*

1) Perform further studies on the analysis of diflunisal to help explain the differences observed in its IC_50_ in vivo and in vitro as this information would make the paper much stronger.

We have conducted additional analysis of diflunisal as suggested. Overexpression of WT p300 suppressed the effect of diflunisal in a dose-dependent manner and increased H2B_K12/K15_ acetylation (new Figure 3), but overexpression of catalytically inactive p300 mutants did not (new Figure 3). Diflunisal also suppressed acetylation of the nonhistone proteins NF-κB p65_K310_ (Figure 3) and p53_K382_ (Figure 3). Here also, Compound C, an AMPK inhibitor, suppressed p-ACC accumulation in response to diflunisal, but did not inhibit deacetylation of acetylated H2B_K12/K15_ or acetyl-p53_K382_, indicating that AMPK is not necessary for these effects (Figure 3). These findings support the model that diflunisal also targets p300 acetyltransferase activity independently of AMPK.

Further, we have observed that salicylic acid/diflunisal metabolism may also contribute to the increased cellular potency observed in vivoin comparison to in vitro. Indeed, we have found that salicyl-CoA, a known major intermediate of salicylate metabolism (Knights, Sykes, and Miners 2007), inhibits CBP/p300 with 28-fold increased potency in comparison with salicylate: IC_50_ =220 µM for salicyl-CoA vs. 6.12 mM for salicylate (new Figure 1—figure supplement 1). A similar 52-fold increase in potency is observed with diflunisal-CoA in comparison to diflunisal: IC_50_ =20 µM for diflunisal-CoA vs. 1.05 mM for diflunisal (new Figure 1—figure supplement 1). Although further investigation will be required to understand the relative contribution of these phenomena to the novel in vivoeffects of salicylate reported here, these observations provide a potential mechanistic basis for the potent cell-based effects of these compounds.

2) Perform further studies to rule out off-target effects perhaps by characterizing one or two of the commonly known targets of diflunisal.

As described above, during the revision of this manuscript, we have conducted additional experiments with overexpressed p300 to demonstrate that the effects of diflunisal are dependent on p300 when examining histone acetylation. We have also tested the effect of p300 overexpression on the anticancer effect of diflunisal on Kasumi cell lines using lentiviral expression vectors for p300 or empty control (new Figure 4). Cells transduced with the empty vector showed inhibition of growth by diflunisal, similar to untransduced cells (new Figure 4). In contrast, p300-transduced cells were significantly more resistant to diflunisal (new Figure 4), and exhibit less apoptosis measured by annexin V positive cells (new Figure 4) and sub-G1 fraction (new Figure 4). These results support the model that diflunisal kills Kasumi-1 cells by apoptosis due to p300 inhibition.

With respect to the suggestion of other possible targets of diflunisal, a review of the literature indicates that the only other reported targets of diflunisal is prostaglandin synthetase, also known as cyclooxygenase (Br J Clin Pharmacol. 1977 Feb; 4 Suppl 1:15S-18S, Proc Natl Acad Sci USA. 1981 Apr; 78(35): 2053-6.). In contrast to aspirin/acetylsalicylic acid, diflunisal does not covalently modify prostaglandin synthetase and probably acts at a site similar to aspirin since diflunisal inhibits acetylation of the enzyme by aspirin (Br J Clin Pharmacol. 1977 Feb; 4 Suppl 1:15S-18S).

The identification of p300 and CBP as novel targets for diflunisal and salicylic acid in addition to cyclooxygenase suggest that diflunisal and salicylic acid could have important novel activities such as inhibition of p300-dependent tumors, as we report here.

*Reviewer #1:*

*In this manuscript, Shirakawa and coworkers characterized salicylate and diflunisal as novel CBP/p300 inhibitors. This finding is surprising but also important for several reasons. First, salicylate is widely known for its anti-inflammatory property by targeting mainly COX1/2 and other proteins. The finding that this molecule also acts on CBP/p300 adds another functionally important target to appreciate the role of salicylate. Second, the Nimer lab previously showed the importance of CBP/p300 for the survival of ETO AML and herein demonstrated that salicylate and diflunisal are effective against ETO AML at the dose relevant to depletion of several acetylation marks. The therapeutic implication of the two compounds is important given their ready use, although the two compounds may hit multiple other targets at the present high doses.*

*The main concerns of the reviewer are:*

*1) Although the role of salicylate against CBP/p300's acetylation activity was rigorously examined by the key rescue experiment shown in Figure 2, diflunisal hasn't been characterized to the same degree. It seems that the authors solely based on the observation that diflunisal is structurally similar to salicylate and then concluded that the two compounds act via the same modes of mechanism. The experiments in Figure 1 and in particular 2B, C and D should be performed for diflunisal as well.*

As requested by the reviewer, we repeated the p300-overexpression rescue experiments using diflunisal and found that overexpression of p300 also suppressed diflunisal-mediated histone H2B deacetylation as shown in our revised new Figure 3 and Figure 3—figure supplement 2. These results support the model that diflunisal also works through p300 inhibition.

*2) For the efficiency of the two compounds against ETO AML, it is not clear whether the effect is achieved by targeting CBP/p300 alone or other targets alone or multiple targets in a combined manner. If the latter are the case, the flow of the work will be disrupted and the impact of the work will be significantly dampened (the potential target(s) of the therapeutic outcome of the two compounds were not CBP/p300).*

The cell line Kasumi-1’s dependency on p300 for tumorigenesis has been well characterized by the Nimer lab (Science. 2011 Aug 5; 333(6043): 765-9). We have tested whether other NSAIDs, including acetaminophen and indomethacin, also inhibit p300 acetyltransferase in vitro, but did not detect any inhibitory activity (not shown). Diflunisal is a known COX-1/2 inhibitor with IC_50_ of 113µM (Proc Natl Acad Sci USA96 (14): 7563-8), but other NSAIDs that lack p300 inhibitory activity failed to inhibit Kasumi-1 cells growth (data not shown). Further, we have also tested the effect of p300 overexpression on the anticancer effect of diflunisal on Kasumi cell lines using lentiviral expression vectors for p300 or empty control (new Figure 4). Cells transduced with the empty vector showed inhibition of growth by diflunisal, similar to untransduced cells (new Figure 4). In contrast, p300-transduced cells were significantly more resistant to diflunisal (new Figure 4), and exhibit less apoptosis measured by annexin V positive cells (new Figure 4) and sub-G1 fraction (new Figure 4). These results collectively support the model that diflunisal inhibits Kasumi-1 cells due to p300 inhibition.

The reviewer strongly feels that the rescue experiments with wild-type and catalytically dead p300 should be carried out at a cellular level in parallel with the experiments in Figure 4 (required) and even better for the animal experiments in Figure 5.

As we described above, we have performed these experiments and found that overexpression of p300 but not catalytically inactive mutants reduced diflunisal-mediated histone H2B deacetylation (new Figure 3 and Figure 3—figure supplement 2). We also performed p300 rescue experiment using leukemia cells and found that p300 transduced Kasumi-1 cells are resistant to diflunisal treatment (new Figure 4). These results support the model that diflunisal exerts its activities (inhibition of histone acetylation and antitumor effect) via p300 inhibition.

*Reviewer #2:*

*Shirakawa et al. assess salicylate derivatives as potential p300/CBP HAT inhibitors based on the fact that other recently developed HAT inhibitors contain a salicylic acid moiety. They first show that salicylate inhibits p300/CBP Hat activity in vitro with an IC_50_ in the 10 mM range. They show that they can inhibit some, but not other histone modifications that are known to be a result of p300/CBP in cells. They go on to identify the small molecule diflunisal as a Hat inhibitor and show that it can inhibit acetylation of AML1-ETO and growth in mice. The data are of interest but there are significant issues regarding specificity of mechanism.*

*1) The* IC_50_*of 10mM for salicylate seems high. Can the authors put this into some context as to the concentrations at which salicylate works against other targets?*

Most of the NSAIDs inhibit both COX-1 and COX-2, although they vary in their relative potencies against the two isozymes (Patrignani, P., Panara, M. R., Sciulli, M. G., Santini, G., Renda, G., and Patrono, C. (1997) J. Physiol. Pharmacol. 48, 623–631). However, salicylate does not, unlike its acetylated derivative aspirin, inhibit COX-1 and COX-2 activity in vitro (Vane, J. R. (1971) Nature New Biol. 231, 232–235; Vargaftig, B. B. (1978) J. Pharm. Pharmacol. 30, 101– 104; Proc Natl Acad Sci USA. 1993 Dec 15; 90(24):11693-7; Cromlish, W.A., and Kennedy, B.P. (1996) Biochem.Pharmacol. 52, 1777–1785). However, salicylate has been shown to exert a comparable analgesic and anti-inflammatory action as aspirin. Salicylates have been proposed to exert their pharmacological effects via inhibition of the transcription factor nuclear factor NF-κB and other targets. In these experimental systems, the concentrations used were in the same range as used in the experiments described in our paper (5-20 mM) (Kopp, E., and Ghosh, S. (1994) Science265, 956–959 17; Pierce, J. W., Read, M. A., Ding, H., Luscinskas, F. W., and Collins, T. (1996) J. Immunol. 156, 3961–3969 18; Schwenger, P., Bellosta, P., Vietor, I., Basilico, C., Skolnik, E. Y., and Vilcek, J. (1997) Proc. Natl. Acad. Sci. USA.94, 2869 –2873; Oeth, P., and Mackman, N. (1995) Blood86, 4144–4152 40; Schwenger, P., Skolnik, E. Y., and Vilcek, J. (1996) J. Biol. Chem. 271, 8089 – 8094 41. Schwenger, P., Alpert, D., Skolnik, E. Y., and Vilcek, J. (1999) J. Cell. Physiol179, 109–114).

It should also be noted that salicylate plasma concentrations in patients taking salicylic acid (3-4 g/day) range between 1-3 mM, a concentration at which partial inhibition of p300/CBP is observed. In addition, our inhibition data in cell lines show a lower IC_50_ (IC_50_ ~5mM for H2B_K12/15_) and new data show that a salicylic acid metabolite, salicyl-CoA inhibits CBP/p300 with 28-fold increased potency in comparison with salicylate: IC_50_ =220 μM for salicyl-CoA vs 6.12 mM for salicylate (new Figure 1—figure supplement 1). These data are consistent with the proposed model that partial or complete inhibition of p300 by salicylate represents one of its relevant biological targets. It is important to note that a small molecule the size of salicylic acid used at such high concentrations is expected to interact with a number of cellular proteins. The observations presented in this manuscript support the hypothesis that some of the biological activities of salicylic acid and diflunisal are mediated via p300 inhibition.

*2) The fact that salicylate inhibits acetylation of specific histone residues is interesting, but the fact that not all p300/CBP dependent modifications are inhibited is concerning in terms of mechanism. The authors explain this by potential redundant enzymes. Anything that can be done here to help explain this result would be very helpful.*

In Figure 2 and Figure 3, we identified histone AcH3K56 as the most sensitive histone acetyl mark to inhibition by both diflunisal and salicylate. This result is highly consistent with the literature showing that the histone acetyl transferase CBP (also known as Nejire) in flies and CBP and p300 in humans acetylate H3K56 (Das C, Lucia MS, Hansen KC, Tyler JK. CBP/p300-mediated acetylation of histone H3 on lysine 56. Nature. 2009 May 7; 459 (7243): 113-7). We also note that the pattern of histone marks inhibition are remarkably similar between the two drugs, but at different concentrations in agreement with their relative abilities to inhibit p300/CBP in vitro. In terms of what is observed for the other sites, the situation is more complex. Indeed, many histone modifications are known to be regulated by a variety of HAT enzymes. For example, acetyl H4_K5_ is regulated by HAT1, CBP, p300, Tip60, and HB01 whereas acetyl H3 K14 is regulated by CBP, p300, PCAF, gcn5, and ScSAS3 (Cell128 (35): 693-705.). We interpret the observed lack of inhibition of histone H3 or H4 acetylation by salicylate or diflunisal to reflect the compensating activities or other histone acetyltransferases that target the same sites.

*3) A significant concern in Figure 3 is that the in vitro* IC_50_*for diflunisal is that the in vitro* IC_50_*is 1mM and the cellular half-maximal inhibition is 100µM. Can this be explained? It seems strange that the* IC_50_*in cells would be less than the* IC_50_*in vitro.*

We think that two distinct mechanism account for this difference. First, as we have discussed in detail in our response to reviewer #1 above, we have repeatedly observed that both salicylic acid and diflunisal treatment lead to the degradation of p300 (Figure 2, Figure 3) indicating that the drugs work both by direct inhibition of the enzyme and drug-induced target degradation. Second, we have also observed that the diflunisal-CoA shows a 52-fold increase in potency in comparison to diflunisal: IC_50_ =20 µM for diflunisal-CoA vs. 1.05 mM for diflunisal (Figure 1—figure supplement 1). We speculate that these mechanisms might explain the discrepancy.

4) The data on inhibition of AML1-ETO cell growth in vitro and in vivo is of interest, however there still remains a specificity issue. The authors need to test growth inhibition on other cell lines without AML1_ETO rearrangement or show that indeed inhibition of AML1-ETO acetylation is the reason for the antiproliferative effect. Also, there needs to be assessment of inhibition of acetylation in the leukemia cells in vivo.

As we discuss above, Kasumi cells dependency on p300 for tumorigenesis has been well characterized by the Nimer lab (Science. 2011 Aug 5; 333 (6043): 765-9). We have tested whether other NSAIDs, including acetaminophen and indomethacin, also inhibit p300 acetyltransferase in vitro, but did not detect any inhibitory activity (not shown). Diflunisal is a known COX-1/2 inhibitor with IC_50_ of 113µM (Proc Natl Acad Sci USA96 (13): 7563-8), but other NSAIDs that lack p300 inhibitory activity failed to inhibit Kasumi-1 cells growth (data not shown). Further, we have also tested the effect of p300 overexpression on the anticancer effect of diflunisal on Kasumi cell lines using lentiviral expression vectors for p300 or empty control (new Figure 4). Cells transduced with the empty vector showed inhibition of growth by diflunisal, similar to untransduced cells (new Figure 4). In contrast, p300-transduced cells were significantly more resistant to diflunisal (new Figure 4), and exhibit less apoptosis measured by annexin V positive cells (new Figure 4) and sub-G1 fraction (new Figure 4). These results collectively support the model that diflunisal inhibits Kasumi-1 cells due to p300 inhibition.

*Reviewer #3:*

The IC_50_ of salicylate for p300/CBP seems to be too high to be specific for HAT/KATs. The authors should show the specificity, using in vivo acetylation of nonhistone substrate, such as p53 acetylation by p300/CBP and PCAF (specific commercial antibodies are available).

As suggested by the reviewer, we have examined the effect of salicylate and diflunisal on a non-histone target and show in Figure 2 and Figure 3 that p53 acetylation is suppressed by both salicylate and diflunisal.

Is it really inhibiting KAT activity in the animal system? The authors may consider to feed the animal (mice) and do immunohistochemistry (IHC) analysis of liver tissue.

Inhibition of cancer progress has been claimed by the authors. The data is quite convincing. Is it possible to show that it is due the inhibition of p300/CBP KAT activity by salicylate?

Direct demonstration that inhibition of p300/CBP KAT activity in the context of tumor growth in vivo would have required the generation of multiple clonal cell lines overexpressing p300 and control cell lines with the empty vector, their characterization (for stability of p300 expression etc…) and a large number of xenograft experiments showing that overexpression of p300 in vivo suppresses the ability of diflunisal to inhibit tumor growth. Given the short time allowed for resubmission, we elected to conduct this experiment on Kasumi cells in vitro. The results presented in our revised Figure 4 show that p300 overexpression increases cell line growth in vitro and suppresses the ability of diflunisal to inhibit cell growth. These important experiments support the model that p300 is critical for the inhibitory effect of diflunisal on the growth of Kasumi cells.

[Editors' note: further revisions were requested prior to acceptance, as described below.]

Reviewer #1:

The revised paper addressed the most of the previous concerns. In particular, the new model of the formation of "salicyl-CoA" and "diflunisal-CoA", if they are indeed the cases, may explain many conflicting data in vitro. However, for such situations, the in vitro biochemical data may not be relevant anymore to evaluate these compounds because they will act differently upon the formation of CoA-adduct. In contrast, the weak interaction of salicyl or diflunisal only argues that HATs contain binding pockets to accommodate these moieties. The reviewer strongly suggests revising the wording of "salicyl or diflunisal as inhibitors of HATs)" into "HATs contain the binding pockets of salicyl or diflunisal to engage the corresponding interaction in vitro and in cellular contexts". By this way, it leaves the possibility that salicyl or diflunisal may be subject to the formation of Co-A adduct and thus act as pro-drugs inside cells. The paper can even be further strengthened by showing the formation of "salicyl-CoA" and "diflunisal-CoA" in a cellular setting.

We have modified the title of the manuscript in the following manner: “Salicylate, diflunisal *and their metabolites* inhibit CBP/p300 and exhibit anticancer activity”. This change acknowledges our observation that the metabolites also exhibit inhibitory activity against p300 and CBP and that we cannot distinguish whether the primary compound or the metabolite is important in the cellular context.

Reviewer #2:

*The manuscript is improved as a result of the new experiments provided. However, while the authors respond to all of the concerns by the reviewers, there is no evidence that these discussions made it into the manuscript. It would be helpful if the authors added some discussion in the manuscript that focus on the reviewers’ questions given it is likely readers will have the same questions.*

*Also, it is not clear why the authors did not expand their assessment of cell lines beyond Kasumi/AML1-ETO. This is not difficult and thus makes one wonder if the compound it toxic to many cell lines.*

We have added new discussion points in the manuscript as recommended by the reviewer.